# Anti-Idiotypic Agonistic Antibodies: Candidates for the Role of Universal Remedy

**DOI:** 10.3390/antib9020019

**Published:** 2020-05-28

**Authors:** Aliya K. Stanova, Varvara A. Ryabkova, Sergei V. Tillib, Vladimir J. Utekhin, Leonid P. Churilov, Yehuda Shoenfeld

**Affiliations:** 1Department of Pathology and Laboratory of the Mosaic of Autoimmunity, Saint Petersburg State University, 199034 Saint Petersburg, Russia; aliya.stanova@mail.ru (A.K.S.); varvara-ryabkova@yandex.ru (V.A.R.); utekhin44@mail.ru (V.J.U.); shoenfel@post.tau.ac.il (Y.S.); 2Department of Immunology, Faculty of Biology, Lomonosov Moscow State University, 119991 Moscow, Russia; tillib@genebiology.ru; 3Institute of Gene Biology, Russian Academy of Sciences, 119334 Moscow, Russia; 4Saint Petersburg Research Institute of Phthisiopulmonology, 191036 Saint Petersburg, Russia; 5Zabludowicz Center for Autoimmune Diseases, Sheba Medical Center, affiliated to Tel Aviv University School of Medicine, Tel-Hashomer, Ramat Gan 52621, Israel

**Keywords:** anti-idiotypic antibodies, agonistic autoantibodies, autacoid, drug, hormone, neurotransmitter

## Abstract

Anti-idiotypic antibodies (anti-IDs) were discovered at the very beginning of the 20th century and have attracted attention of researchers for many years. Nowadays, there are five known types of anti-IDs: α, β, γ, ε, and δ. Due to the ability of internal-image anti-IDs to compete with an antigen for binding to antibody and to alter the biologic activity of an antigen, anti-IDs have become a target in the search for new treatments of autoimmune illnesses, cancer, and some other diseases. In this review, we summarize the data about anti-IDs that mimic the structural and functional properties of some bioregulators (autacoids, neurotransmitters, hormones, xenobiotics, and drugs) and evaluate their possible medical applications. The immune system is potentially able to reproduce or at least alter the effects of any biologically active endogenous or exogenous immunogenic agent via the anti-idiotypic principle, and probably regulates a broad spectrum of cell functions in the body, being a kind of universal remedy or immunacea, by analogy to the legendary ancient goddess of universal healing Panacea (Πανάκεια, Panakeia in Greek) in the treatment and prevention of diseases, possibly including non-infectious somatic and even hereditary ones.

## 1. Introduction

In 1900, London and Besredka demonstrated the existence of physiologic antihemolysins. In fact, these were anti-idiotypic autoantibodies (anti-IDs) against hemolysins. The authors immediately suggested their regulatory role [1]. Much later experimental anti-IDs were obtained by Kryzhanovsky et al. (1960) [2] and by Oudin (1963) [3]. In 1973, Lindenmann speculated that some anti-IDs may serve as homobodies or internal immunological mirror images of appropriate antigens [4]. The interest in anti-IDs started to increase with the proposal of the network theory of the immune system by Jerne in 1974 [5]. The main idea was that Ab can both recognize an antigen and in turn also be recognized by another Ab towards its paratope (Figure 1).

The mechanism of the formation of Jerne’s network is the following: An antigen stimulates the production of Abs (Ab1). The active centers of Ab1 are recognized by the second class of Abs—anti-IDs (Ab2). In their turn, Ab2 serves as an antigen for the third class of Abs (Ab3)—anti-anti-IDs, and so on [6]. The network finally appeared to be not endless, because Ab3 may be identical or close in their recognizing properties to Ab1. The system is self-balanced and follows thermodynamically Le Châtelier’s principle of shift compensation (Le Châtelier′s principle states that if a dynamic equilibrium is disturbed by changing the conditions, the position of equilibrium shifts to counteract the change to reestablish equilibrium), hence the appearance of any new antigenic specificity “X“ or an increase in the amount of any existing Ab tends to make the system respond by increasing the production of anti-X- or anti-Ab specificities. In this interpretation, the immune system does not care about the self- or non-self-character of recognizable antigens, and rather keeps the idiotype–anti-idiotypic balance. Moreover, according to Jerne’s theory it is nothing but autoimmunity, which serves as the key point of the physiologic autoregulation of the immune system.

Anti-IDs are directed against the idiotype of Ab1 and may represent its mirror replica. It seems that some of them can recognize the same structures as the antigen does. Paul Ehrlich was the first who pointed out in Latin: “*Corpora non facit nisi fixata*”—“*The bodies cannot act by other means than to bind*” [7]. The recognition of the same structures both by an antigen and by its anti-IDs in living systems may elicit similar informational and signaling consequences. If anti-IDs do recognize and bind like their primary antigens, they might to a certain degree act as the antigen, and imitate or competitively block some (or even all) biological effects of primary antigens (hormones, xenobiotics, neurotransmitters, autacoids, and drugs). This option gives an overwhelming hope that the immune system can reproduce any immunogenic molecule, partially or entirely mimicking the activity of such bioregulators and even drugs.

So, spontaneously arising anti-IDs may have a therapeutic effect through the mechanism presented in Figure 2, and earlier we named this potency of the immune system, the effect of Immunacea, associating the term with the ancient all-healing goddess Panacea [8]. The discussion about the possible mechanisms of intravenous immunoglobulin’s (IVIG’s) therapeutic action in various diseases [9,10] shall take into account the probable role of anti-IDs imitating the remedies ever taken by their donors.

Due to the ability of internal-image anti-IDs to compete with an antigen for binding to Ab1 and to alter the biologic activity of an antigen, anti-IDs have become a target in the search for new treatments of autoimmune illnesses, cancer, and some other diseases. Additionally, three autoimmune diseases were experimentally induced [11] by means of idiotypic immunization of the mice (systemic lupus erythematosus—SLE, anti-phospholipid syndrome—APLS, and Wegener’s granulomatosis—WG). In the case of WG, mice normally do not express the target autoantigen (neutrophil proteinase 3 enzyme), but anti-IDs can substitute the missing signal in the pathogenesis of the disease.

In this review, we summarized the data on anti-IDs, which mimic the structural and functional properties of some bioregulators, and to evaluate their possible medical applications.

## 2. Classes of Anti-IDs

Nowadays, there are five known types of Ab2s recognized: α, β, γ, ε, and δ. Ab2α binds to an idiotope of Ab1, which does not take part in antigen recognition, and therefore Ab2α cannot inhibit binding of the antigen to Ab1. Ab2β recognizes an idiotope of Ab1, which belongs to the paratope, involved in antigen recognition. Ab2β is able to inhibit antigen binding to Ab1, and also able to stimulate the production of Ab3. These two categories of Ab2s are defined by Bona and Kohler (1984) [12]. Ab2γ has similar properties to Ab2β but cannot induce Ab3 production. Ab2ε, or epibody, has an ability to bind not only to Ab1 but to the antigen. The newest type of Ab2 is Ab2δ, which recognizes a non-binding site within the variable region of the heavy chains [13,14,15].

From these five types of Ab2s mentioned, the anti-idiotypic antibodies of Ab2β type have attracted special attention. The antigen-competing properties of these Ab2β allow one to suggest that the paratopes of particular Ab2β can resemble the antigenic epitope and thus may carry ‘internal images’ of it. Hence, this type of Ab2 can be employed as a surrogate antigen. However, structural analyses have shown that the mimicking of antigenic epitopes is functional and mediated by similar binding interactions but does not depend on the close identity of their 3-D structures [16]. An essential detail is that nonstructural but functional mimicry can even be observed with anti-idiotypic antibodies reacting with Ab1 to non-protein antigens [17,18,19].

## 3. Autacoid-Like Anti-IDs

Under the term “autacoids” (from Greek: “autos” (self) and “acos” (relief, drug)), the biomedical thesaurus means the bioregulators of short distant local and zonal wireless action, produced, acting, and metabolized topically, for example, within inflammatory foci or in lymphoid follicles [20].

To be autacoids means to have a certain way and zone of action, not belonging to any one particular chemical group. Many cytokines, biogenic amines, eicosanoids, and polysaccharides, as well as purines or nitric oxide belong to this type of bioregulator and act both in health and disease.

There are several reports on the generation of anti-IDs that were able to mimic cytokine effects. Monoclonal anti-ID AA1E5 was generated [21], which contained an internal image of human interferon-γ (IFN-γ). AA1E5 binds to the receptor of human IFN-γ, and together with its Fv fragment reproduces the antiviral activity of human IFN-γ. Interleukin-1 (IL-1) is a proinflammatory cytokine and is one of the signals required to induce T-cell proliferation. It was demonstrated [22] that the rabbits, immunized with IL-1, produced anti-IDs, which were similar in their bioactivity to IL-1. To prove this, two independent in vitro assays were performed. The results showed that, like IL-1, anti-IDs enhanced the proliferation of thymocytes and Con A-stimulated D10 cells.

Another remarkable study demonstrated that anti-idiotypic antibodies from a rabbit immunized with anti-angiotensin monoclonal antibody could bind to rat liver membranes bearing angiotensin receptors, proving the ability of anti-IDs to mimic different molecules, both structurally and functionally [23]. In human studies, the agonistic AA against angiotensin II type I receptor was strongly associated with pre-eclampsia in pregnancy [24].

The other evidence for anti-IDs’ participation in the regulation of the normal immune response was shown [25]. The authors examined the effect of rabbit anti-IDs against human IgG $F{\left(ab\prime \right)}_{2}$ anti-tetanus toxoid (TT) antibodies—on the synthesis of TT-specific immunoglobulin E (IgE) by peripheral blood lymphocytes in vitro. This study demonstrated that IgE antibody synthesis is controlled by anti-IDs not only in non-allergic but also in allergic subjects. It was shown [26] that anti-IDs to basophil-bound idiotypes, which mimic the antigen - Lolium perenne (perennial rye grass) pollen allergen (Lol p I), can alter histamine release from basophils.

## 4. Neurotransmitter-Like Anti-IDs

A neurotransmitter is a chemical bioregulator of distant wire-dependent topical action, acting within the synapses between the producing and target cells. One of the noteworthy experiments with anti-IDs altering the neurotransmission was that performed by Elazar et.al. [27], who obtained monoclonal anti-IDs against antibodies (Abs) to haloperidol. Those anti-IDs mimicked the action of haloperidol on the D-2 dopamine receptors and successfully bound to them.

Polyclonal rabbit anti-IDs against monoclonal anti-dopamine antibodies were produced [28] and inhibited the binding of both polyclonal and monoclonal idiotypic anti-dopamine antibodies directed towards immobilized anti-dopamine conjugates. It was also shown that anti-IDs inhibit the binding of tritium-matched anti-dopamine ligand to rat brain membranes. The last experiment proved that anti-IDs could cross-react with a peptide extracted from a neuroblastoma cell line (NCB-20), known to express functional dopamine receptors.

Anti-IDs with acetylcholine-receptor-like ligand-binding properties were isolated from the sera of four patients with myasthenia gravis [29]. In another study, anti-IDs were purified and demonstrated binding abilities similar to various cholinergic ligands. In the next research, anti-IDs were generated by immunizing rabbits with Abs to serotonin (5-hydroxytryptamine; 5-HT) [30]. The results showed that those anti-IDs recognized the 5-HT1B, 5-HT1C, and 5-HT2 receptor subtypes, whereas the 5-HT1A subtype was not recognized by anti-IDs.

There are many reports about autoantibodies (AAs) against neurotransmitter receptors with agonistic activity occurring in patients. We assume an anti-idiotypic origin of such AAs, because they activate the receptor and trigger the signal cascades similarly to the method that natural agonists use. The findings of a recent study provide support to the concept that AA against the receptors of neurotransmitters (among AA against other G-protein-coupled receptors) are natural components of human biology [31]. The authors showed that these AAs are present across multiple chronic diseases as well as in the sera of healthy donors, and form network signatures, which are dependent on such factors as age, gender, the presence of the disease, and its nature. The most common theories of AA production, which are based on molecular mimicry and immune dysregulation, are unable to fully explain the occurrence of such a wide spectrum of natural IgG, as was observed in this study. Jerne’s network theory can help to explain the abundance of these AAs since in the state of the idiotype–anti-idiotypic balance, the production of AAs appears to be not so heavily restricted. Regarding the functional properties of antineurotransmitter AAs, it was found [32] that AAs against β2-adrenergic receptors may display agonistic activity. Moreover, there is strong evidence that these AAs are associated with macroangiopathy in cases of longstanding diabetes mellitus type 2. It was also demonstrated [33] that agonistic AAs against the β2-adrenergic receptor are associated with the development of primary open-angle glaucoma (POAG). The agonistic AAs against the β1-adrenergic receptor rendered cardiopathogenic effects [34]. Another group revealed [35] agonistic AAs against α-1-adrenoceptor in approximately 50% of patients with Alzheimer’s disease. The patients with chronic fatigue syndrome had elevated titers of AA both against the β2-adrenergic receptor and muscarinic acetylcholine receptors [36].

## 5. Hormone-Like Anti-IDs

Hormones “sensu stricto” are chemical bioregulators of distant wireless action. Promising results came from the studies of anti-IDs mimicking the bioeffects of hormones. An early experiment [37] was performed by using rat Abs specific for bovine insulin as an immunogen for inducing anti-IDs in rabbit. Surprisingly, a fraction of these anti-IDs were able to inhibit the binding of insulin to its receptor. In addition, the binding of anti-IDs to insulin receptors on thymocytes functionally mimicked the binding of insulin as assessed by the uptake of α-amino-isobutyric acid. The ability of a monoclonal anti-ID (1D5) directed against the binding site of a monoclonal anti-estradiol Ab to interact with the estrogen receptor (ER) was investigated [38]. The results showed that 1D5 has the capacity to mimic the bioactivity of estradiol, recognizing ER. Monoclonal anti-ID B-32 to growth hormone was developed [39]. B-32 specifically interacted with growth hormone receptors expressed on target cells, activated growth hormone receptor and Janus kinase [2] signal transducers, as well as the JAK2/STAT5 signaling pathways. In addition, B32 could also stimulate porcine hepatocytes to secrete insulin-like growth factor-1. So, B-32 serves as a growth hormone receptor agonist mimicking the effects of growth hormone.

The use of anti-IDs in immunotherapeutic strategies is very promising, and some beneficial results were reached. It was shown [40] that the administration of anti-IDs for human anti-glutamate decarboxylase 65 mAb to non-obese diabetic (NOD) mice delayed the onset of the experimental sugar diabetes and reduced the severity of underlying autoimmune insulitis. Comparing human, mouse, and guinea pig anti-IDs to insulin [41], the researchers found that there is a species-wide cross-reactivity between anti-insulin idiotypes. Anti-IDs towards the insulin receptor were able to render a short-time stimulation effect on receptors, with their subsequent desensitization.

Similar anti-IDs were revealed in patients in many cases of “cryptogenic” hypoglycemia; they are common for the insulin resistance of B-type and were registered both in patients who received insulin injections and in those who had never used exogenous insulin [42]. Several authors [42,43] highlighted the occurrence of anti-insulin receptor AA, which binds to the insulin receptor, mimics insulin action, and causes fasting hypoglycemia. There are some reports about patients (especially, those of Japanese ethnic origin), who developed spontaneous anti-insulin AA causing hypoglycemic attacks.

The researchers [44] called the condition accompanied by the production of such AA the insulin autoimmune syndrome or Hirata’s disease. The characteristics of insulin autoimmune syndrome are the combination of fasting hypoglycemia, a high concentration of total immunoreactive insulin, and the presence of AA to native human insulin in sera. The insulin-mimicking AA in these cases could be of anti-idiotypic origin. The agonistic AA towards the insulin receptor were also found in several systemic autoimmune diseases, like SLE [45,46,47].

The sum of such experiments and clinical data supported the idea that the anti-ID network may operate in humans to generate functionally active agonistic AA to hormone receptors. It was previously demonstrated [48] that mice immunized against bovine insulin developed anti-IDs, which in vitro imitated in assays some properties of Abs to insulin receptors. The development of anti-ID anti-receptor antibodies appeared to be associated with the insulin resistance of adipocytes and abnormalities of glucose homeostasis.

Several reports about the prolactin-like activity of anti-prolactin receptor Abs were published. It was observed [49] that anti-prolactin receptor Abs at moderate concentrations reproduce the prolactin effect on casein gene expression and DNA synthesis in a culture of rabbit mammary explants. The effect of anti-prolactin receptor Abs on prolactin binding sites on the rat liver cells was investigated [50]. At lower concentrations, these Abs mimic the upregulatory effect of prolactin, which means that the prolactin molecule itself is not permanently required for its bioeffects beyond its initial binding to receptors. In vivo effects of anti-prolactin receptor Abs administered to pseudo-pregnant rabbits were demonstrated [51]. Similar to previous studies, this one also found the prolactin-like activity of anti-prolactin receptor Abs both in vitro and in vivo. When injected to pseudo-pregnant rabbits, Abs against the prolactin receptor induced an accumulation of β-casein or stimulation of β-casein biosynthesis and an increase in its mRNA concentration in breast cells. The effects of anti-prolactin receptor antibodies in nitrosomethylurea-induced mammary tumors in organ cultures were investigated [52]. It was demonstrated that anti-prolactin receptor Abs were able to induce an increase in lactose synthetase activity, DNA synthesis, and prolactin-binding sites, thus mimicking the prolactin actions in rat mammary cells. Because of the well-known stimulating activity of prolactin over the autoimmune processes [53], the prolactin-like effects of anti-IDs may link into the pathogenesis of hyperprolactinemic autoimmune disorders as a kind of vicious circle.

Graves’ disease (GD) or diffuse toxic goiter is caused by Abs against the thyroid-stimulating hormone (TSH) receptor [54]. The occurrence of Abs against *Yersinia enterocolitica* (YE) was registered in patients with GD [55,56]. It was demonstrated [57] that TSH receptor autoantibodies bind to YE-specific binding sites, similar to the TSH receptor antigens of thyroid membranes. Many studies showed that GD is caused by autoimmunity against the TSH receptor, which in turn is caused by poorly regulated immunity against YE. One of the first suggestions about the key role of anti-IDs in GD was coined [58]. In a series of experiments [59,60,61], the authors developed human anti-thyrotropin anti-ID. These anti-IDs inhibited the binding of TSH to thyroid plasma membranes and stimulated their adenylate cyclase activity. It was proposed that anti-TSH receptor Abs that cause GD may elicit anti-IDs against thyrotropin.

## 6. Anti-IDs to Xenobiotics

More than 20 years ago, researchers started to produce anti-IDs to different xenobiotics and toxins. Monoclonal anti-ID against HEC toxin, produced by *Aeromonas hydrophila*, was presented [62]. Anti-IDs bound rabbit polyclonal antibodies (PAb) against HEC toxin in the Dot-ELISA test and inhibited the binding of PAb to HEC toxin in a blocked ELISA. It was shown that these anti-IDs might serve as a surrogate antigen and induce specific immune reactivity to the natural microbial antigen. Monoclonal anti-ID that mimics the mycotoxin of dothistromin (DOTH) was prepared [63]. One anti-ID antibody, 12C9G8, was equivalent to DOTH on a molar basis in the inhibition of the binding of an anti-DOTH mAb to DOTH–protein conjugates. Furthermore, in a Western blot analysis, 12C9G8 visualized the same protein profile in embryos as was observed in an approach that utilized an anti-DOTH mAb together with DOTH-mouse serum albumin conjugate to identify potential DOTH binding sites. Single chain fragment (ScFv) anti-IDs of a killer toxin (KT) from the yeast *Pichia anomala* have been produced by recombinant DNA methodology from the splenic lymphocytes of mice immunized by idiotypic vaccination with a KT-neutralizing mAb KT4 [64]. ScFv KT-like anti-IDs react with specific *Candida albicans* KT cell wall receptors (KTRs) exerting a fungicidal activity in vitro, which could be neutralized by adsorption with mAb KT4. More importantly, ScFv KTIdAb displayed an effective therapeutic activity in an experimental rat model of vaginal thrush.

Anti-IDs show promising results in controlling the hyperergic immune response towards various allergens. The anti-ID IgG Fab-fragments, mimicking the *Phleum pratense* grass pollen allergen Phl p 5, were obtained [65]. It was produced using phage display technology from a human allergic donor, following selection against human polyclonal anti-allergen IgE. Other authors vaccinated mice suffering from induced acute allergic bronchial asthma with these anti-IDs. Results showed that in treated animals, the extent of acute eosinophilic inflammation upon rechallenge with aerosolized allergen was lower than in untreated asthmatic and non-asthmatic ones [66].

## 7. Anti-IDs to Drugs and Treatment of Drug Addiction

The anti-alprenolol anti-IDs, which bind to the β-adrenergic receptors, were produced [67]. These anti-IDs stimulated the basal adenylate cyclase activity and modulated the catecholamine signal transmission to the catalytic unit. The experiment was performed on turkey erythrocytes. The importance of these data is related to the fact of the association between clinically found $\beta_{1}$-adrenergic receptor AA and human cardiovascular and systemic diseases, like postural tachycardia and other arrhythmias, chronic fatigue syndrome, etc. [34,68].

One of the most intriguing applications of anti-ID antibodies is the development of treatments against drug addiction, exploiting the anti-ID principle. Anti-IDs, which specifically recognize and bind opiate receptors, were developed [69]. Anti-IDs inhibited the binding of naloxone to mouse brain homogenate. Anti-IDs that behave like morphine molecule were also created [70]. The authors showed that anti-IDs inhibited the binding of morphine to opiate receptors and to anti-morphine Abs.

Two monoclonal anti-IDs, which interacted with the binding site of opioid receptors, were generated [71]. The monoclonal anti-β-endorphin Ab served as an antigen. Results showed that anti-IDs interacted both with membrane-bound and solubilized opiate receptors of the µ- and δ-types. Moreover, anti-IDs demonstrated opiate antagonistic activity by their ability to reduce the action of DADL ([D-Ala^2^, D-Leu^5^]-enkephalin) on cyclic adenosine monophosphate (cAMP) production in neuroblastoma x glioma cell line. Roy et al. [72] revealed in IVIG samples naturally occurring human anti-IDs to anti-β-endorphin with the property of binding to opiate receptors. They showed that anti-ID inhibited the binding of idiotypic anti-β-endorphin IgG to β-endorphin. Moreover, anti-ID inhibited the binding of β-endorphin to the opiate receptor. Other researchers isolated human anti-δ-opiate receptor (DOR) autoantibodies from IVIG by affinity purification [73]. Autoantibodies were able to bind specifically to DOR. Moreover, these autoantibodies were agonistic as seen by their ability to inhibit forskolin-stimulated cyclic adenosine monophosphate (cAMP) accumulation, as well as receptor desensitization via phosphorylation of DOR. IVIG contains purified IgG isolated from the plasmas of thousands of apparently healthy donors.

We suppose that such natural anti-ID towards opiates could be elicited in some of the multiple IVIG donors, who used opiates in the parenteral route, medically or non-medically.

The researchers [74] produced monoclonal an anti-ID antibody K2-3f, which mimicked a cocaine molecule and bound hDAT with a higher affinity than cocaine itself. More importantly, K2-3f at a low molar concentration completely inhibited cocaine binding to hDAT while simultaneously allowing equimolar dopamine uptake.

According to the idea of Immunacea [8], immunization with Abs against ligand, which is absent or severely deficient in the organism, may cause the production by lymphocytes with more or less precise immunological copies of a missing ligand. In fact, the reality of this phenomenon has already been demonstrated in the abovementioned experiments with WG modeling in mice [11]. Thus, we presume that immunization with opiates (or other objects of drug addiction) may open an anti-idiotypic way for the prevention and treatment of opiate (or other) addiction. Since such low molecular weight molecules as drugs of abuse are not immunogenic, the traditional way to create anti-addiction vaccines is to render the immunogenicity of the drug molecules thorough chemical modification enabling covalent linkage to protein. Nonetheless, despite multiple immunizations, none of the vaccines developed to date resulted in high titers of high-affinity anti-drug antibodies in a majority of patients [75]. Exploiting the anti-IDs principle is an alternative way to induce the immune response against the drug of abuse. However, there is a paucity of literature on this issue [76,77]. The most recent study, published in 2019, deals with the development of anti-idiotypic-based vaccines against various morphine derivatives [77]. Similarly, we presume that if there is a genetically determined lack of a certain protein in the organism, probably the immunization with the Abs against missing protein may force lymphocytes to produce its anti-Id substitute, which would promise an immunologic way for the treatment of some genetic defects.

There are different methods for developing hormone-like antibodies. One of them is the immunization of animals with antisense peptides. Antisense RNAs may encode the peptides recognizing each other. For example, ACTH (adrenocorticotropic)-like antibody was obtained via the immunization of mice with complementary antisense peptide for ACTH, named HTCA [78]. The monoclonal antibody, developed by this method, showed ACTH-like activity in adrenocortical cell culture. Depending on the concentrations, it either stimulated isolated rat adrenal cells to secrete corticosterone or inhibited mitotic activity in mouse Y-1 adrenal cells. Earlier, other ways to produce antibodies with hormonal activity were shown. Polyclonal rabbit antibodies able to reproduce some effects of ACTH on the post-receptor level were raised by immunization with deoxyribonucleoprotein of the rat adrenal cortex. The effect was cell specific and non-reproduced in adrenals neither by preimmune IgG nor by anti-hepatic ones [79]. Monoclonal antibodies against the ACTH receptor, a few of them with agonistic properties, were developed by the immunization of mice with porcine adrenal cortex cell membranes [80].

## 8. Anti-IDs in the Pathogenesis of Diseases

An approach based on anti-IDs is quite attractive as a strategy for the treatment of different autoimmune diseases. It was demonstrated [81] that the vaccination of nine humans with SLE with lupus autoantibodies in a phase 1 trial induced the development of anti-ID in five patients, who remained disease free for the two-year follow-up period. The anti-IDs against monoclonal rheumatoid factor (Ka m-RF) were developed by a cell fusion procedure [82]. In vitro, these anti-IDs strongly suppressed the production of rheumatoid factor by lymphocytes from unrelated rheumatoid arthritis patients with cross-reactive idiotypes. These results indicate that anti-ID antibody may influence the regulation of rheumatoid factor production in patients with rheumatoid arthritis. It was observed [83] that NOD mice, which are prone to autoimmune type 1 diabetes mellitus (T1D), develop high titers of anti-ID after the injection of a human GAD65Ab monoclonal Ab. Since these animals did not develop T1D and had reduced pathomorphological signs of insulitis, the induction of anti-ID was linked to the prevention of T1D in this animal model. The breathtaking observation was made that the lack of anti-IDs to GAD65Ab distinguished in humans the diagnosis of T1D [84]. The authors revealed the presence of these anti-IDs in the majority of healthy individuals, whereas in patients with T1D, the amount of GAD65Ab-specific anti-IDs was severely reduced. The experiment, which revealed the presence of anti-acetylcholine receptor anti-IDs in the sera of patients with myasthenia gravis, was performed [85]. Another group of researchers [86] isolated FVIII inhibitor-specific anti-IDs from phage-displayed libraries. Those anti-IDs successfully blocked the binding of FVIII inhibitors to FVIII and neutralized in vitro the activity of the inhibitors.

We suppose that in the future, similar anti-IDs can be used to restore FVIII deficiency in patients with hereditary hemophilia A, in accordance with the abovementioned Immunacea principle.

Promising results were achieved in the development of anti-IDs for the treatment of some infectious diseases. Several successful studies in mice included the induction of protection against *Trypanosoma* [87], *Escherichia coli* [88], and *Streptococcus pneumoniae* [89]. It was demonstrated [90] that an anti-ID, which mimics an oligosaccharide epitope from *Neisseria gonorrhoeae*, was more effective than the nominal antigen in inducing bactericidal antibodies in mice and rabbits. Another example of Ab2 mimicking the chlamydial glycolipid exoantigen was provided [91]. Mice immunized with an anti-ID were protected against infection with *Chlamydia trachomatis*.

## 9. Conclusions

The data on the biological effects of anti-IDs received from experiments are summarized in Table 1.

All these studies connected with anti-IDs proved their potential as a powerful tool not only in immunological assays but also in the treatment of various diseases. Apparently, these are biological instruments active not only within the immune system but also beyond it, when targeted to somatic cells. For example, IDs have been exploited as therapeutic immunogens in cancer treatment in two well-defined and clearly distinct contexts: (i) Directly as a tumor-specific target on membrane Ig-positive malignant B cells as a consequence of their clonotypic origin, and (ii) as a surrogate of tumor-associated antigen (TAA) to induce specific immune responses [92]. However, although several animal studies using anti-Id antibodies support their utility as cancer vaccines, human trials with monoclonal Ab2β were disappointing and have failed in later phase trials [93]. Reasons for the failure of anti-Id vaccines against tumors are similar to the generalized failures of other cancer vaccines. One of the major problems in cancer is the complexity and heterogeneity of antigen expression, as the antigens that are potential targets of T- and B-cells are multiple and endlessly adaptable. However, this obstacle could be overcome. The experience of polyclonal anti-ID-based reagents in animal models as well as an understanding of the immune response in humans lends to the proposition that polyclonal anti-ID vaccines will be more effective compared to monoclonal-based ones [93]. It should be mentioned that there are some difficulties in generating anti-Ids themselves, and the main one is that the ID of Ig is nominally a self-antigen and consequently poorly immunogenic. The employment of adjuvants, carriers, viral vectors, or the direct recruitment of immune-related cells are the methods used to overcome this problem [92]. We also suggest that the discovery of single-domain antibody fragments (VHHs or Nanobodies^®^) with a high stability and solubility in camelids, which already have certain therapeutic applications [94,95], would alleviate the difficulties in the production of anti-IDs for the treatment of various human diseases. Future investigations of agonistic anti-IDs may entirely bring an idea of Immunocea to life. We hypothesize that not only agents of the immune cells but all the bioregulators and their receptors in the body (both cell surface and intranuclear ones) physiologically are included in the network of idiotypic–anti-idiotypic interactions and in autorecognition. Probably, this is the basis for physiologic immunoglobulin-mediated regulation of cell functions and growth, not only in disease but in health also. Future medicine shall harness these mechanisms for the treatment and prevention of illnesses [79,96]. Some work in the area of medical application of anti-IDs has already been started. Anti-IDs, which mimic the function of different catalytic molecules (abzymes), have been obtained [97]. Their therapeutic potential (including, for example, neutralization of toxic substances in the treatment of chemical poisoning) is under investigation [98]. Anti-IDs against bioregulators, such as autacoids, neurotransmitters, hormones, xenobiotics, and drugs, should not be excluded, as the same approach is applicable to these anti-IDs.

## Figures and Tables

**Figure 1 antibodies-09-00019-f001:**
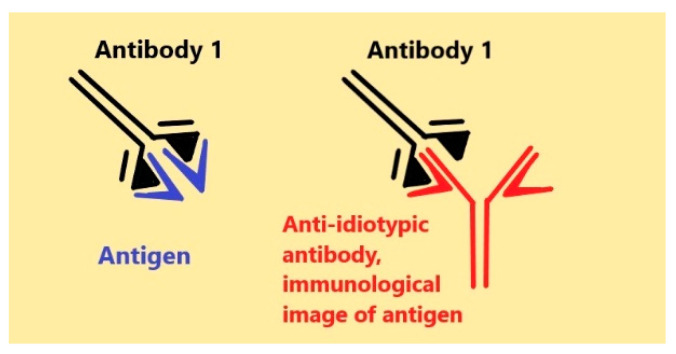
Scheme of idiotypic/anti-idiotypic interactions.

**Figure 2 antibodies-09-00019-f002:**
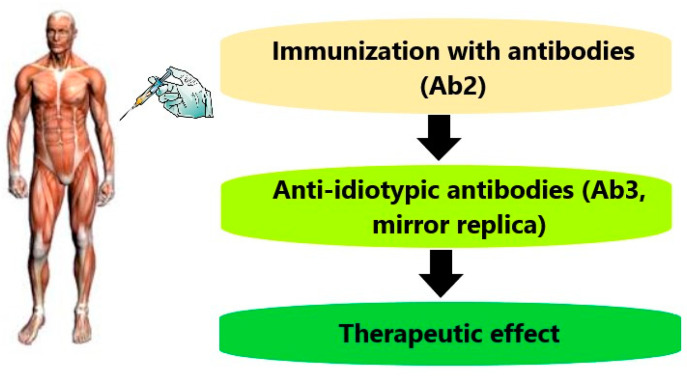
Proposed mechanism of anti-idiotypic antibodies therapeutic action.

**Table 1 antibodies-09-00019-t001:** Experiments with anti-IDs mimicking bioregulators.

Anti-ID Bearing Internal Image of …	Biological Effects of Anti-ID	Authors
Human interferon-γ (IFN-γ)	Binding to the receptor of HuIFN-γ and reproduces antiviral activity	Depraetere et al. [21]
Interleukin-1 (IL-1)	Enhancing proliferation of thymocytes	Zuberi et al. [22]
Angiotensin	Binding to angiotensin receptors	Couraud et al. [23]
Basophil-bound idiotype	Altering histamine release from basophils	Mecheri et al. [26]
Haloperidol	Binding to D-2 dopamine receptors	Elazar et al. [27]
Dopamine	Inhibiting binding of dopamine to rat brain membranes	Mons et al. [28]
Acethylcholine-receptors	Similar binding abilities for cholinergic ligands	Eng et al. [29]
Serotonin (5-hydroxytryptamine, 5-HT)	Recognizing 5-HT1B, 5-HT1C and 5-HT2 receptor subtypes	Tamir et al. [30]
β2-adrenergic receptor	Agonistic activity	Werner et al. [32] Junemann et al. [33]
β1-adrenergic receptor	Agonistic activity	Bornholz et al. [34]
α1-adrenergic receptor	Agonistic activity	Karczewski et al. [35]
Insulin	Binding to insulin receptors on thymocytes	Sege et al. [37] Elias et al. [41] Taylor et al. [42] Chong et al. [43]
Estradiol	Recognizing of estrogen receptor	Mor et al. [38]
Growth hormone	Binding to growth hormone receptors, stimulation of porcine hepatocytes to secrete insulin-like growth factor-1	Lan et al. [39]
Prolactin	Prolactin-like activity on casein gene expression and DNA synthesis	Djiane et al. [49] Dusanter-Fourt et al. [51]
Thyrotropin	Inhibition biding of thyroid stimulating hormone to thyroid plasma membranes and stimulating their adenylate cyclase activity	Farid et al. [58,59,60,61]
ACTH	Stimulation of isolated adrenal cells to secrete corticosterone, inhibition of mitotic activity in adrenal cells.	Clarke et al. [78]
HEC toxin of *Aeromonas hydrophila*	Induction of specific immune reactivity to natural microbial antigen	Chen et al. [62]
Mycotoxin of dothistromin	Binding to dothistromin-protein conjugates	Jones et al. [63]
Killer toxin (KT) of *Pichia anomala*	Reaction with KT receptors, therapeutic activity in experimental rat model of vaginal thrush	Magliani et al. [64]
*Phleum pretense* grass pollen allergen	Therapeutic effect on experimental mice model of acute allergic asthma	Hantusch et al. [65]
Alprenalol	Binding to β-adrenergic receptors, stimulating of basal adenylate cyclase activity, modulation of catecholamine signal transmission	Schreiber et al. [67]
Naloxone	Binding to opiate receptors	Ng et al. [69]
Morphine	Inhibition of the morphine binding to opiate receptors	Glasel et al. [70]
β-endorphin	Opiate antagonistic activity	Gramsch et al. [71] Roy et al. [72]
Cocaine	Binding to hDAT	Ho et al. [74]

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
