# Peer review of "Anti-Idiotypic Agonistic Antibodies: Candidates for the Role of Universal Remedy"

_2073-4468, 2020, doi:10.3390/antib9020019_

Round 1

Reviewer 1 Report

In the manuscript “Anti-idiotypic agonistic antibodies: candidates for 2 the role of Immunacea”, the authors summarize the data about anti-IDs that mimic structural and functional properties of some bioregulators as autacoids, neurotransmitters, hormones, xenobiotics, and drugs. The review is very well structured according to this analysis.

One of the most intriguing applications of anti-ID antibodies is the development of treatment against drug addiction, exploiting the anti-ID principle. However, the references from this topic cited by the authors are relatively old. The newest is from 2011. It would be of advantage to shortly describe the state-of-the-art in this field.

This review lacks comments about the difficulties in generating antiIDs. In the interest of the reader, it would be good to consider them at least in general terms.

I recommend the authors to consider removing the term “Immunacea” from the title. This term is not widely known. Its obscurity can push the reader away from this interesting article, which can decrease the review‘s impact.

Author contributions are missing.

The reviewer recommends accepting this manuscript with minor corrections.

Author Response

Thank you for your letter and for the opportunity to revise our paper “Anti-idiotypic agonistic antibodies: candidates for 2 the role of Immunacea”. The suggestions offered by the Reviewers have been immensely helpful, and we also appreciate your insightful comments. They were incorporated into the revised version of the manuscript: clear and with highlighted changes. We hope, that you will find the corrected version feasible for publication in Antibodies.

Point 1: One of the most intriguing applications of anti-ID antibodies is the development of treatment against drug addiction, exploiting the anti-ID principle. However, the references from this topic cited by the authors are relatively old. The newest is from 2011. It would be of advantage to shortly describe the state-of-the-art in this field.

Response 1: We agree with the Reviewer, but the recent literature on this issue in English is very scarce. We add the comment about it and cite the most recent paper (2019) which deals with the anti-ID principle  in the development of vaccines against drugs of abuse. Line 300: “Since such  low molecular weight molecules as drugs of abuse are not immunogenic, the traditional way to create anti-addiction vaccines is to render the immunogenicity of the drugs molecules thorough chemical modification enabling covalent linkage to protein. Nonetheless, despite multiple immunizations, none of the vaccines developed to date resulted in high titers of high affinity anti-drug antibodies in a majority of patients(75). Exploiting the anti-IDs principle is an alternative way to induce the immune response against drug of abuse. However, there is a paucity of literature on this issue(76,77). The most recent study, published in 2019, deals with the development of anti-idiotypic based vaccines against various morphine derivatives(77)”
Point 2: This review lacks comments about the difficulties in generating antiIDs. In the interest of the reader, it would be good to consider them at least in general terms.

Response 2: We agree with the Reviewer, and cited the recent paper, which address this issue and consider the main difficulties in general terms. Line 376: “It should be mentioned, that there are some difficulties in generating anti-Ids themselves, and the main one is that the ID of Ig is nominally a self-antigen and consequently poorly immunogenic.The employment of adjuvants, carriers, viral vectors, or the direct recruitment of immune-related cells are the methods used to overcome this problem(91). We also suggest that the discovery of single-domain antibody fragments (VHHs or Nanobodies®) with a high stability and solubility in camelids, which already have certain therapeutic applications(93,94), would alleviate the difficulties in production anti-IDs for the treatment of various human diseases. “

Point 3: I recommend the authors to consider removing the term “Immunacea” from the title. This term is not widely known. Its obscurity can push the reader away from this interesting article, which can decrease the review‘s impact.

Response 3: We agree with the Reviewer and removed the term “Immunaea” from the title.

Point 4: Author contributions are missing.

Response 4: We added Author contributions. Line : “Author Contributions: Conceptualization, L.P.; Investigation, A.S., V.R., S.T., V.U.; Writing – Original Draft Preparation, A.S.; Writing – Review & Editing, V.R., Y.S., L.P.; Visualization, A.S.; Supervision, Y.S..; Funding Acquisition, L.P., Y.S. Concept of Immunacea, L.P.; Section 3, V.U., Section 7, S.T., Section 8, L.P.”

Reviewer 2 Report

The present paper is a review focused on anti-idiotypic antibodies and their possible usefulness in the treatment of various human pathologies. The authors describe the evidences suggesting these antibodies as tools for the prevention and treatment of chronic diseases such as allergy, cardiovascular diseases and cancer, and infectious diseases.

Broad comments

The topic addressed is interesting and the authors cite studies conducted in animal models and humans related to a wide spectrum of human pathologies indicating these antibodies as a kind of Immunacea. To provide a more complete overview, the authors should also cite studies demonstrating a failure in the therapeutic use of these antibodies. In particular, they should cite the review published in 2019 by Kohler H et al. (Kohler H, Pashov A, Kieber-Emmons T. The Promise of Anti-idiotype Revisited. Front Immunol. 2019 Apr 12;10:808.).

They also should revise the manuscript to improve English language.

Specific comments

The authors should quote Figure 2 in line 67 and not in line 63 (in this line therapeutic effects are not mentioned).

Line 119: the reference 36 is not correct. Please, insert the right one.

Line 257: please, insert reference. The reference 64 cited in the previous sentence is not related to studies in mice.

Author Response

Thank you for your letter and for the opportunity to revise our paper “Anti-idiotypic agonistic antibodies: candidates for 2 the role of Immunacea”. The suggestions offered by the Reviewers have been immensely helpful, and we also appreciate your insightful comments. They were incorporated into the revised version of the manuscript: clear and with highlighted changes. We hope, that you will find the corrected version feasible for publication in Antibodies.

Point 1: To provide a more complete overview, the authors should also cite studies demonstrating a failure in the therapeutic use of these antibodies. In particular, they should cite the review published in 2019 by Kohler H et al. (Kohler H, Pashov A, Kieber-Emmons T. The Promise of Anti-idiotype Revisited. Front Immunol. 2019 Apr 12;10:808.).

Response 1: We agree with the reviewer, and cited the review published in 2019 by Kohler H et al. However, this review deals only with the therapeutic use of monoclonal anti-ID-based cancer vaccines, so not all therapeutic applications, discussed in our paper are considered by Kohler H.

See in the revised version of the manuscript: Line 362: “For example, IDs have been exploited as therapeutic immunogens in cancer treatment in two well-defined and clearly distinct contexts: (i) directly as a tumor-specific target on membrane Ig-positive malignant B cells as a consequence of their clonotypic origin, and (ii) as surrogate of tumor-associated antigen (TAA) to induce specific immune responses(91). However, although several animal studies using anti-Id antibodies support their utility as cancer vaccines, human trials with monoclonal Ab2β were disappointing and have failed in later phase trials(92). Reasons for the failure of anti-Id vaccines against tumors are similar to generalized failures of other cancer vaccines. One of the major problems in cancer is the complexity and heterogeneity of antigen expression, the antigens that are potential targets of T and B-cells are multiple and endlessly adaptable. However, this obstacle could be overcome. The experience of polyclonal anti-ID-based reagents in animal models as well as an understanding of the immune response in humans lends to the proposition that polyclonal anti-ID vaccines will be more effective compared to monoclonal-based ones(92).”

Point 2: They also should revise the manuscript to improve English language.

 Response 2: We agree with the Reviewer and revised the manuscript to improve English language (use of articles, word order, style, grammar etc).

Point 3: The authors should quote Figure 2 in line 67 and not in line 63 n this line therapeutic effects are not mentioned).

Response 3: We agree with the reviewer and quoted Figure 2 in the appropriate line.

Point 4: Line 119: the reference 36 is not correct. Please, insert the right one.

Response 4: We agree with the Reviewer and inserted the right reference  - (24). (Xia Y, Zhou CC, Ramin SM, Kellems RE. Angiotensin Receptors, Autoimmunity, and Preeclampsia. J Immunol. 2007 Sep 15;179(6):3391–5.)

Point 5: Line 257: please, insert reference. The reference 64 cited in the previous sentence is not related to studies in mice.

Response 5: We agree with the Reviewer and inserted reference  - (66) (Wallmann J, Epstein MM, Singh P, Brunner R, Szalai K, El-Housseiny L, et al. Mimotope vaccination for therapy of allergic asthma: Anti-inflammatory effects in a mouse model. Clin Exp Allergy. 2010 Apr;40(4):650–8.)
